# Using Unoccupied Aerial Systems (UASs) to Determine the Distribution Patterns of Tamanend's Bottlenose Dolphins (*Tursiops erebennus*) across Varying Salinities in Charleston, South Carolina

Nicole Principe [1,*] , Wayne McFee [2], Norman Levine [3], Brian Balmer [4] and Joseph Ballenger [5]

1 Graduate Program in Marine Biology, College of Charleston, Charleston, SC 29424, USA
2 National Oceanic and Atmospheric Administration, National Ocean Service, National Centers for Coastal Ocean Science, Charleston, SC 29412, USA; wayne.mcfee@noaa.gov
3 Department of Geology and Environmental Geosciences, College of Charleston, Charleston, SC 29424, USA; levinen@cofc.edu
4 Dolphin Relief and Research, Clancy, MT 59634, USA
5 South Carolina Department of Natural Resources, Charleston, SC 29412, USA; ballengerj@dnr.sc.gov
* Correspondence: nicole.principe@noaa.gov

**Abstract:** The Charleston Estuarine System Stock (CESS) of Tamanend's bottlenose dolphins (*Tursiops erebennus*) exhibit long-term site fidelity to the Charleston Harbor, and the Ashley, Cooper, and Wando Rivers in Charleston, South Carolina, USA. In the Cooper River, dolphins have been irregularly sighted in upper regions where salinity levels are below what is considered preferred dolphin habitat. We conducted unoccupied aerial system (UAS) surveys in high-salinity (>15 parts per thousand) and low-salinity (<15 parts per thousand) regions (*n* = 8 sites) of the Cooper River and surrounding waters to assess dolphin distribution in terms presence/absence, detection rate, abundance, and density. We also assessed the influence of ecological factors (salinity, water temperature, season, and prey availability) on dolphin distribution. Dolphins were detected at five sites, with higher salinity and water temperature being significant predictors of presence and abundance. Dolphins were detected year-round across high-salinity sites, and were infrequently detected in low-salinity sites during months with warmer water temperatures. The results from this study contribute to the overall understanding of dolphin distribution across various habitats within the Charleston Estuary System and the potential drivers for their movement into low-salinity waters.

**Keywords:** drones; aerial surveys; marine mammals; estuary; environmental factors

## 1. Introduction

Along the east coast of the United States, Tamanend's bottlenose dolphins (*Tursiops erebennus*), previously referred to as common bottlenose dolphins (*Tursiops truncatus*) [1], are found in coastal waters, as well as inshore habitats, including bays, sounds, and estuaries [2–4]. The ranging and distribution patterns of bottlenose dolphins can occur across broad-scale geographic regions and fine-scale habitats, and be influenced by various abiotic and biotic factors, including salinity [5,6], water temperature [7,8], tidal cycle [9,10], prey/predator abundance [11,12], or a combination of these and other factors. Dolphins inhabiting estuaries are particularly vulnerable to cumulative stressors as these habitats are highly dynamic in which drastic changes in water temperature and salinity can be physiologically challenging to dolphins [13,14]. Further, estuaries are prone to higher anthropogenic activity than offshore waters, which increases the likelihood of dolphins being exposed to stressors such as vessel strikes [15,16], fishery entanglements [17], point and non-point source contaminants [18], and disease [19]. Therefore, knowledge of

the spatial distribution and habitat preferences of dolphins within estuarine environments is necessary for continued management and conservation efforts.

The Charleston Estuary System (CES) is an ecologically and economically important estuary centered in Charleston Harbor, South Carolina, USA. Charleston Harbor receives saltwater influx from the Atlantic Ocean and freshwater input from three main river systems: the Ashley, Cooper, and Wando Rivers. The long-term population monitoring of bottlenose dolphins in the CES began in 1994, which led to the identification of three distinct groups based on migratory patterns: year-round residents, seasonal residents, and transients [20]. Photo identification (photo ID) studies from 2003 to 2007 estimated the population size of dolphins within the CES to be between 364 and 910 animals, with the highest abundances occurring during the summer [21]. Excluding transient animals, the best population estimate for resident dolphins is between 289 and 319 individuals [21,22]. Resident dolphins are classified as a distinct stock known as the Charleston Estuarine System Stock (CESS). The distribution of CESS dolphins varies across sex and season with the entrance to the Charleston Harbor being a large core use area for males and females [22]. While resident CESS dolphins are the primary focus of this study, coastal dolphins from the Western North Atlantic South Carolina/Georgia Coastal Stock also have ranges extending into the larger water bodies of the study area (e.g., Folly River, Charleston Harbor).

The Cooper River has been the least surveyed area within the geographic boundaries of the CESS. This river extends ~89 km northward from Charleston Harbor [23], with habitat characteristics shifting markedly as the water changes from saline to brackish to freshwater [24]. There are numerous industrial influences on this riverine system, including U.S. Navy port facilities, commercial operations associated with the State Ports Authority, and various private entities [24]. High levels of persistent organic pollutants (POPs) have been identified in sediment and dolphin prey species within the Cooper River [25,26], as well as within dolphins throughout the CES [27]. Significant modifications to flow regimes have also occurred in the Cooper River to harness hydroelectric power, which in turn has impacted freshwater discharge [28,29].

With limited fine-scale data on dolphins within the Cooper River, it is currently unclear how dolphins utilize this riverine system within the broader context of the CES, as well as potential impacts associated with anthropogenic activity and habitat modification to dolphins in this area. Over the past 20 years, there have been opportunistic sightings and intermittent strandings ($n = 11$) of dolphins reported in the northern, low-salinity habitats of the Cooper River, as well as an adjacent freshwater tributary (Back River) [30]. Bottlenose dolphins have evolved in marine and estuarine environments where salinity levels are typically ~30 parts per thousand (ppt). While estuarine populations have been observed in lower salinities (15–30 ppt), there is a low salinity threshold that no longer constitutes preferred dolphin habitat, as prolonged exposures could bring adverse health effects and mortality. Low salinity can be defined as <15 ppt, and previous research delineated 8 ppt to be the minimum threshold for preferred dolphin habitat [5]. Estuarine dolphins may be exposed to brief periods of lower salinity waters due to freshwater runoff from rivers or large storm events (e.g., floods and hurricanes) [10,31]. However, prolonged exposure to freshwater can result in the sloughing of skin and ulcerative lesions, changes in pathophysiology, lobomycosis (the first suspected report of this disease in South Carolina was observed in 2020), and mortality [13,32–34].

Currently, it is unclear at what temporal and spatial scales dolphins are utilizing the lower salinity habitats of the Cooper River. Previous reports of prolonged freshwater exposure in dolphins throughout the southeastern U.S. have primarily been associated with out-of-habitat animals in which individual animals were displaced due to extreme weather events (e.g., hurricanes and floods), or for reasons that are unclear, remained in a freshwater environment for extended periods of time [32,35]. Other hypotheses for dolphins shifting into lower saline environments include sea level rise, providing additional habitat that may not have been previously available [36], or they are following prey or moving into this area as prey concentrations in preferred lower estuary habitats decrease [30,37]. Regardless of

the driver(s) associated with dolphin use in the lower saline environments of the Cooper River, prolonged exposure has the potential for negative health consequences.

Within the last decade, there has been a surge in testing the applicability of using unoccupied aerial systems (UASs) for marine mammal research [38]. UASs (or, commonly, "drones") can alleviate certain challenges associated with other methodologies for studying free-ranging marine mammals (e.g., variable movement patterns, inaccessible habitats, and lack of continuous visibility), as these methodologies can be repeated over time across extended distances, and provide permanent visual, georeferenced records of wildlife sightings [39–41]. UAS methodologies have shown promise for marine mammal detection [42–44], behavioral observations [45], and the photogrammetric assessment of body condition [46,47]. While initial studies demonstrate UASs as being beneficial to marine mammal studies, there remains a need for studies with more robust design and data analyses to ensure reliable and reproducible results [45]. Further, few UAS studies have focused specifically on bottlenose dolphins (e.g., [48]), and none have been conducted in a turbid, complex estuarine environment, as in the CES. For estuarine populations of dolphins that are facing cumulative threats, a UAS has the potential to fill a unique niche in data collection methods and provide insights into the biology and overall health of populations.

The aim of this study was to utilize UAS methodology to assess bottlenose dolphin distribution across high- and low-salinity regions of the CES in terms of (1) presence/absence, (2) detection rates, and (3) local abundance and density. The goal was to understand if dolphin distribution extends upriver into the low-salinity habitat of the Upper-Cooper River (UCR) and Back River, and whether factors such as salinity, water temperature, season, and prey availability influence dolphin presence and abundance across the survey regions. This study also assessed the benefits and limitations of using UASs for studying wild cetacean populations in a complex, estuarine environment. The specific hypotheses being tested within the CES were (1) the presence and abundance of bottlenose dolphins are greater in higher salinity regions than in lower salinity regions, (2) bottlenose dolphin presence and abundance varies seasonally across the survey regions, and (3) environmental factors and prey availability influence dolphin presence and abundance across the survey regions.

## 2. Materials and Methods

### 2.1. Study Area

The CES is a semi-enclosed estuary centered in Charleston Harbor (32.7694° N, −79.8953° W) and includes the main channels and tributaries of the Ashley, Cooper, and Wando Rivers (Figure 1). This estuarine system is surrounded by extensive brackish and salt marsh habitats [23,49]. Two high and low tides (i.e., semi-diurnal tides) occur daily with a mean tidal amplitude of 1.6 m [50]. Water clarity is generally poor in this area due to high productivity, strong tidal currents, and soft substrate [51].

UAS surveys were conducted from land-based study sites (*n* = 8) across a wide gradient of salinities (~0–30 ppt) along the Cooper River and in surrounding estuarine and coastal waters near Charleston, South Carolina. Study sites were grouped into two distinct classifications based upon general salinity classifications (Figure 1).

High-salinity study sites (>15 ppt):

1. Folly River: Interior-side barrier island river that is adjacent to the Atlantic Ocean.
2. Harbor–Cooper River Confluence: Deep, open area estuary confluence where the Charleston Harbor branches into the Cooper and Wando Rivers.
3. Mid-Cooper River: Brackish riverine habitat.

Low-salinity study sites (<15 ppt):

4. Upper-Cooper River (UCR) Site 1: Brackish/freshwater riverine habitat.
5. Upper-Cooper River (UCR) Site 2: Brackish/freshwater riverine habitat.
6. Upper-Cooper River (UCR)/Back River Confluence: Open freshwater confluence where the Back River branches off the Cooper River.
7. Back River Site 1: Freshwater riverine habitat.

8. Back River Site 2: Freshwater riverine habitat; terminal end of the Back River.

**Figure 1.** Land-based study sites (*n* = 8) for unoccupied aerial system (UAS) surveys of bottlenose dolphins within the Charleston Estuary System (CES).

*2.2. UAS Survey Methodology*

UAS surveys of bottlenose dolphins were conducted between January 2021 and January 2022 using one of two DJI quadcopters: (1) a DJI Phantom 4 or (2) a DJI Mavic Air 2 (SZ DJI Technology Co., LTD, Nanshan, Shenzhen, China). Both models have a similar battery life (~20 min) and record video footage in 4K high definition (3840 × 2160 pixels) at 30 frames per second. Surveys were conducted on days when weather was most favorable (i.e., no precipitation, winds < 20 kph/Beaufort Scale 3 or below, visibility > 4.8 km). Flight plans were submitted using the AirMap application (AirMap, Inc., Santa Monica, CA, USA) for each flight following Federal Aviation Administration (FAA) airspace regulations. Surveys were conducted in the morning and early afternoon to target various tidal states and minimize sun glare. Multiple flights (1–5) were completed per day depending on weather conditions, available battery life, and time constraints. A 30 min interval was given between flights to reduce the potential for re-sighting the same animals. Surveys were conducted each month, with a rotating, randomized schedule for specific study sites each week. Some sites were not able to be surveyed every month due to weather and schedule conflicts, and sites in the lower estuary and lower Cooper River were prioritized.

At each study site, the UAS was launched from land from a home point and flown manually at 30 m in altitude along transects no more than 2 km away from the home point (Figure 2). This distance ensured that the UAS remained within a visual line of sight to the remote pilot and allowed the UAS to remain in strong connection to the remote control.

Transects were designed to cover the entire region as much as possible while minimizing flying over the same region twice. For example, the UAS was flown up one side of the river, then flown back on the opposite side. However, the narrowness of certain rivers led to the entire river width being visible in the frame while flying up one side and flying back on the other side. It was generally easy to discern if the same dolphin or group was re-encountered while flying along a different transect based on the group size, behavior, and direction they were heading in. Surveying was defined as 'on-effort' until the entire study site was surveyed or the UAS reached low battery (~30% battery life), at which time it was then flown 'off-effort' back to the home point as the crow flies (i.e., shortest distance back to landing zone) at a max UAS speed of 10 m/s, which was typically <2 min of flight time). No dolphins were recorded during 'off-effort' flying; however, the time spent following dolphins was still considered as "on-effort". Survey effort was determined for each study site using the total duration of flight time (mins). The UAS was maneuvered at a speed of 5–10 m/s with the camera angled between 18 and 22° down from horizontal to minimize sea surface glare and provide a large field of view while scanning for dolphins (NMFS Permit No. 21938-03). In-flight data (i.e., live video feed, battery life, distance, altitude, number of satellites) were monitored via the DJI GO 4 or DJI Fly application (SZ DJI Technology Co., LTD, Nanshan, Shenzhen, China) on an iPhone 10 (Apple Inc., Los Altos, CA, USA) connected to the remote control.

When a dolphin or group was detected, the UAS was maneuvered to approach and follow the animals. The UAS camera was positioned straight down to obtain the clearest view of the dolphins. The goal of each sighting was to obtain video footage to discern group size, composition, and behavior. The UAS was maintained at 30 m in altitude with brief descents to >9 m for fine-scale observations of dolphins (NMFS Permit #21938-03). An individual or group was followed until (1) all necessary data were collected, (2) dolphins were lost or traveled out of range of the UAS, (3) a new individual /group was detected, or (4) the UAS signaled a low battery warning and had to return to the home point. Depending on group size, behavioral activity, and footage capture success, multiple consecutive flights may have been conducted for the same individual/group. High-definition video was recorded for every flight, whether dolphins were observed or not, for subsequent review. Data for each flight were recorded in the field using the ArcGIS Survey123 application (ESRI, Redlands, CA, USA), including date/time, study site, weather (i.e., temperature, precipitation, wind speed/direction, cloud cover), flight duration, and preliminary dolphin data (i.e., presence/absence of dolphins, total number of dolphins, and general behavior).

*2.3. UAS Video Analysis*

Post-survey, all video files were downloaded, and flight records were uploaded to airdata.com. Videos from each flight were extensively reviewed using QuickTime Media Player v 10.7 (Apple Inc., Los Altos, CA, USA) to record dolphin detections. Every dolphin or group encountered by the UAS was considered a detection, with group size and composition recorded for each detection. When the same animal or group was reencountered, it was considered a new detection. However, when the total number of dolphins was summed per day, known repeat sightings of animals were not included. Mom and calf pairs were identified by the appearance of a smaller dolphin swimming tightly in echelon position (i.e., in close proximity with its mother's mid-lateral flank) [52]. Flight records were used to obtain the exact start and end of each flight, total flight time, and obtain the GPS coordinates and time of each dolphin detection.

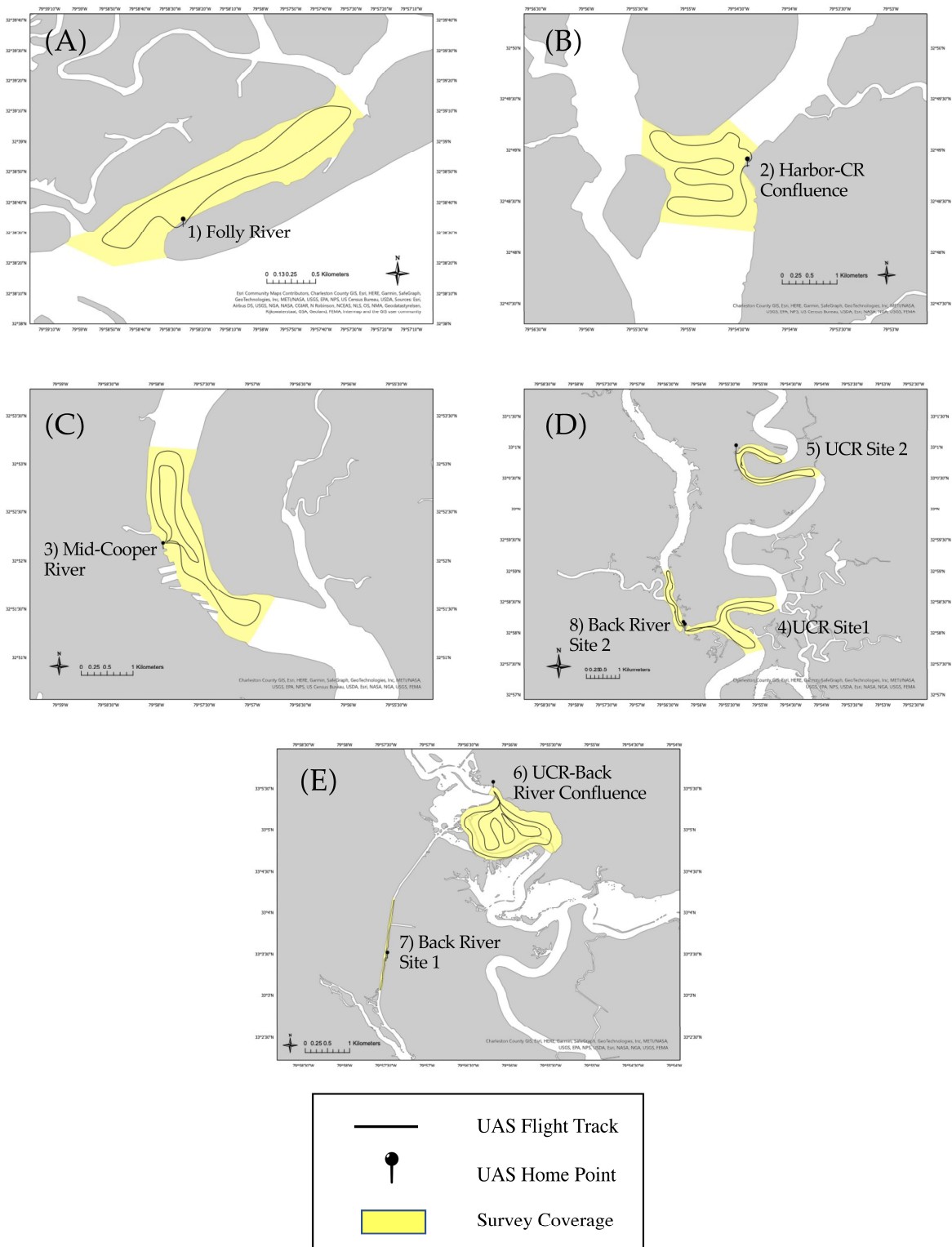

**Figure 2.** UAS survey coverage with flight tracks and home points for all study sites (*n* = 8) within the Charleston Estuary System (CES): (**A**) Folly River, (**B**) Harbor–Cooper River Confluence, (**C**) Mid-Cooper River, (**D**) Upper-Cooper River Sites 1 and 2, Back River Site 2, (**E**) UCR/Back River Confluence, Back River Site 2.

*2.4. Environmental Data Collection*

　　Salinity and water temperature were recorded for each flight from coastal monitoring stations (*n* = 6) closest to each study site (Figure 1). Data were recorded from several

USGS stations [53] and the Southeast Coastal Ocean Observing Regional Association (SECOORA) NOAA National Estuarine Research Reserve System (NERRS) Fort Johnson station (portal.secoora.org). For stations that did not report salinity, specific conductance values (uS/cm at 25 °C) were converted to salinity (parts per thousand) using R code based on a general equation for salinity [54]. Data from the UCR Sites 1 and 2 were combined as they possessed comparable habitat characteristics and shared water quality data from a single monitoring station. Historical salinity data (2000–2021) from the UCR were plotted to examine trends over time (see Supplementary Text S1 for details) [55–57].

### 2.5. Statistical Analysis

#### 2.5.1. Distribution across Study Sites

Distribution across study sites was assessed in terms of (1) presence/absence, (2) detection rate, and (3) abundance and density. For presence/absence, each flight was marked as "yes" if dolphins were detected or "no" if dolphins were not detected. The percentage of flights with dolphins present was calculated across each study site and across all study sites. Daily detection rates (detection/h of flight time) were calculated for each study site:

$$\text{Daily Detection Rate} = \frac{\text{\# of dolphin detections per day}}{\text{survey time (mins) per day}} \times 60 \text{ min}$$

Three out of the eight study sites (UCR–Back River Confluence, Back River Sites 1 and 2) did not have any dolphin detections. To assess whether the daily detection rate was significantly different among the five sites with detections, a Kruskal–Wallis rank sum test was used. If there was a statistical significance reported ($p < 0.05$), a post hoc all pairwise multiple comparison test (Dunn's test) was performed.

For local abundance, the total number of individual dolphins detected at each study site was summed. Flight tracks at each study site were overlayed on ArcGIS and the total area covered (km$^2$) was measured. The total number of dolphins was then divided by the area covered to determine density at each site. To assess whether the number of dolphins significantly differed among study sites, the three low-salinity sites (UCR/BR Confluence, Back River Sites 1 and 2) where no dolphin detections occurred were omitted, and only sites where dolphins were encountered were included. Further, UAS flights that knowingly included the same dolphin or group as a previous flight were not included to reduce the over-estimation of the total number of dolphins seen. A Kruskal–Wallis test and subsequent Dunn's test were performed to assess whether the mean number of dolphins differed between study sites.

A one-way ANOVA was applied to determine whether group size significantly differed amongst study sites. Flights without dolphin detections were not included in this analysis and those that knowingly included the same dolphin or group as a previous flight were also omitted. A log transformation was used on the data to correct for normality. A Tukey's Honest Significant Difference (HSD) test was performed as a post hoc multiple comparison test.

#### 2.5.2. Seasonal Analysis

For analysis of seasonal patterns in dolphin abundance, seasons were defined based upon previous studies using mean temperature: winter (December–February), spring (March–May), summer (June–August), and fall (September–November) [22,58]. For this analysis, data collected in December 2021 were combined with January and February 2021 to represent winter. Additional data collected in January 2022 were excluded from statistical analysis. A Kruskal–Wallis test and subsequent Dunn's test were performed to examine the relationships between dolphin abundance and seasons. Sightings were plotted for each season by group size using ArcGIS Pro Version 3.0 (2022, ESRI, Redlands, CA, USA).

### 2.5.3. Environmental Variables

Daily dolphin counts were plotted against the daily average salinity and water temperature for each study site where dolphins were present. UCR Site 1 and Site 2 were combined as environmental data were collected from the same USGS station. Generalized linear models (GLMs) were used to analyze the relationship between environmental variables and dolphin presence and abundance. GLMs are useful as these models allow for a flexible approach to data analysis and provide an integrated theory of modelling that encompasses the most important models for both continuous and discrete response variables [59]. The salinity and water temperature during each flight were recorded as well as a "yes" if dolphins were present or "no" if dolphins were not present. The presence/absence of dolphins relative to salinity and water temperature was investigated using a binomial GLM with a logit-link function. A second, focused binomial GLM was run only including the two upper estuarine sites where dolphins were sighted (UCR Sites 1 and 2). In the UCR sites 1 and 2, salinity fluctuated anywhere from 0 to 12 ppt. By running a more focused GLM at these two sites, the goal was to investigate whether dolphins were utilizing these areas more at times when salinity was at its highest and was, therefore, a more suitable habitat. The odds ratio for each predictor variable was calculated using the following formula:

$$e^{\beta}, \text{ where } \beta = \text{regression coefficient estimate}$$

The 95% confidence interval for the odds ratio of each predictor variable was calculated using the following formula:

$$e^{(\beta +/- 1.96 * \text{std error})}$$

Total number of dolphins were fitted using a zero-inflated negative binomial (ZINB) regression model to assess the possible effects of salinity and water temperature on dolphin abundance across all study sites. The ZINB model was chosen as a preliminary analyses suggested both overdispersion (choose negative binomial over Poisson error distribution) and zero-inflation existed in the dataset. Zero-inflated models are two-component mixture models capable of dealing with excess zero counts by combining a count distribution (i.e., negative binomial) with a point density at zero [60]. Excess zeros commonly occur in field-collected data as observation records may include false zeros as a result of survey detection errors (i.e., a dolphin was present but not detected, or an individual was absent from a suitable habitat, leading to an excessively high frequency of zeros in the data) [61]. The excess zero results in complicated inferences and incorrect assumptions of species−habitat associations were not accounted for [62]. Zero-inflated models provide a better fit as they account for both true (unsuitable habitat) and false zero observations [63]. The zero-inflated models help account for 'false zeros' (i.e., failure to detect an individual that is indeed present) due to variation in detection ability due to weather and other environmental conditions (i.e., high turbidity). Factors influencing detectability are further described in the discussion. A zero-inflated model thus produces two models, one for the count and one for excess zeros, essentially determining which predictors affect the probability of a dolphin being present and which predictors affect how many dolphins are present.

The results were considered statistically significant at $p < 0.05$. The odds ratios and 95% confidence intervals for each significant predictor variable were calculated using the same formulas as above. All analyses were completed using R Statistical Software (v4.0.2) [64].

### 2.5.4. Prey Data Collection and Analysis

Fish data were obtained from electrofishing surveys conducted by the South Carolina Department of Natural Resources (SCDNR) Inshore Fisheries Department from May 2001 to August 2022. A subset of data was analyzed from surveys conducted across 56 stations in a small subset of the UCR (Figure 1).

Stomach content analysis from dolphins stranded in the UCR/Back River habitat identified several species, including Atlantic croaker (*Micropogonias undulatus*), silver perch (*Bairdiella chrysoura*), spotted seatrout (*Cynoscion nebulosus*), bay anchovy (*Anchoa mitchilli*),

snook (*Centropomus undecimalis*), and spot (*Leiostomus xanthurus)* as prey items [30]. Additional species present in the UCR that have been identified as prey for dolphins in South Carolina include American eel (*Anguilla rostrata*), pinfish (*Lagodon rhomboides*), red drum (*Sciaenops ocellatus)*, southern flounder (*Paralichthys lethostigma*), and striped mullet (*Mugil cephalus*) [65]. Catch per unit effort (CPUE) of these 11 species was calculated for each electrofishing survey:

$$\text{CPUE} = \frac{\text{number of fish caught}}{\text{duration of survey (mins)}} \times 15 \text{ min}$$

The CPUE of these 11 species were plotted separately for the winter, spring, summer, and fall of 2021.

## 3. Results

### 3.1. UAS Survey Effort

A total of 296 UAS flights (~83 h of flight time) were conducted across eight study sites within the CES between January 2021 and January 2022 to survey for bottlenose dolphins (Table 1). The number of flights and total survey time were comparable between high-salinity study sites (*n* = 148 flights; 2569 min total flight time) and low-salinity study sites (*n* = 148 flights; 2429 min total flight time). The mean flight duration ranged from 15 to 18 min (max. 26 min). The UAS was launched and operated from a land-based home point; therefore, the amount of area that could be covered for each site was limited.

**Table 1.** Survey effort from unoccupied aerial system (UAS) surveys of bottlenose dolphins across land-based study sites (*n* = 8) within the Charleston Estuary System (CES).

| Study Site | Total Months Surveyed | Total Flights Conducted | Total Flight Time (mins) | Area Covered (km$^2$) | Temporal Data Gaps |
|---|---|---|---|---|---|
| Folly River (1) | 12 | 44 | 729.13 | 1.3 | August |
| Harbor–CR Confluence (2) | 13 | 55 | 944.77 | 2.2 | None |
| Mid-Cooper River (3) | 13 | 49 | 895.1 | 2.3 | None |
| Upper-Cooper River Site 1 (4) | 13 | 48 | 840.9 | 2.7 | None |
| Upper-Cooper River Site 2 (5) | 13 | 32 | 544.21 | 1.4 | None |
| UCR–Back River Confluence (6) | 9 | 21 | 332.26 | 1.5 | January 2021/2022, July, August |
| Back River Site 1 (7) | 11 | 20 | 307.62 | 0.2 | July, January 2022 |
| Back River Site 2 (8) | 13 | 27 | 404.04 | 0.5 | None |

### 3.2. Dolphin Distribution across Study Sites

Bottlenose dolphins (*n* = 189) were detected (i.e., present) in 26% of UAS surveys (Figure 3, Table 2). Dolphins were only detected at five out of the eight study sites, thus reducing the overall detection rate. Dolphins were sighted at all high-salinity sites (Folly River, Harbor–Cooper River Confluence, Mid-Cooper River) and only two low-salinity sites (UCR Site 1 and UCR Site 2). The daily detection rate (number of detections per hour of flight time) differed across these five sites ($\chi2$ = 66.6, *df* = 7, *p* < 0.001), with rates generally decreasing as study sites moved upriver (Figure 4). The results from the Dunn's test found significant differences between the Folly River and UCR Sites 1 and 2, the Harbor–Cooper River Confluence and UCR Sites 1 and 2, and the Mid-Cooper River and UCR Sites 1 and 2. The Harbor–Cooper River Confluence had the highest number of dolphin detections (*n* = 34) and overall number of dolphins (*n* = 70), while the UCR Site 2 had the lowest (*n* = 3 detections; 5 dolphins; Table 2). The mean number of individual dolphins detected also differed across study sites ($\chi2$ = 30.37, *df* = 4, *p* < 0.001), with the Dunn's test finding significant differences between the Folly River and UCR Sites 1 and 2, the Harbor–Cooper River Confluence and UCR Sites 1 and 2, and the Mid-Cooper River and UCR Sites 1 and 2. The group size of dolphins did not differ across study sites (ANOVA, $F_{4,63}$ = 0.42, *p* > 0.05).

Calves were detected at the five study sites, but were more commonly observed in the Folly River and Harbor–Cooper River Confluence (*n* = 8, 7, respectively).

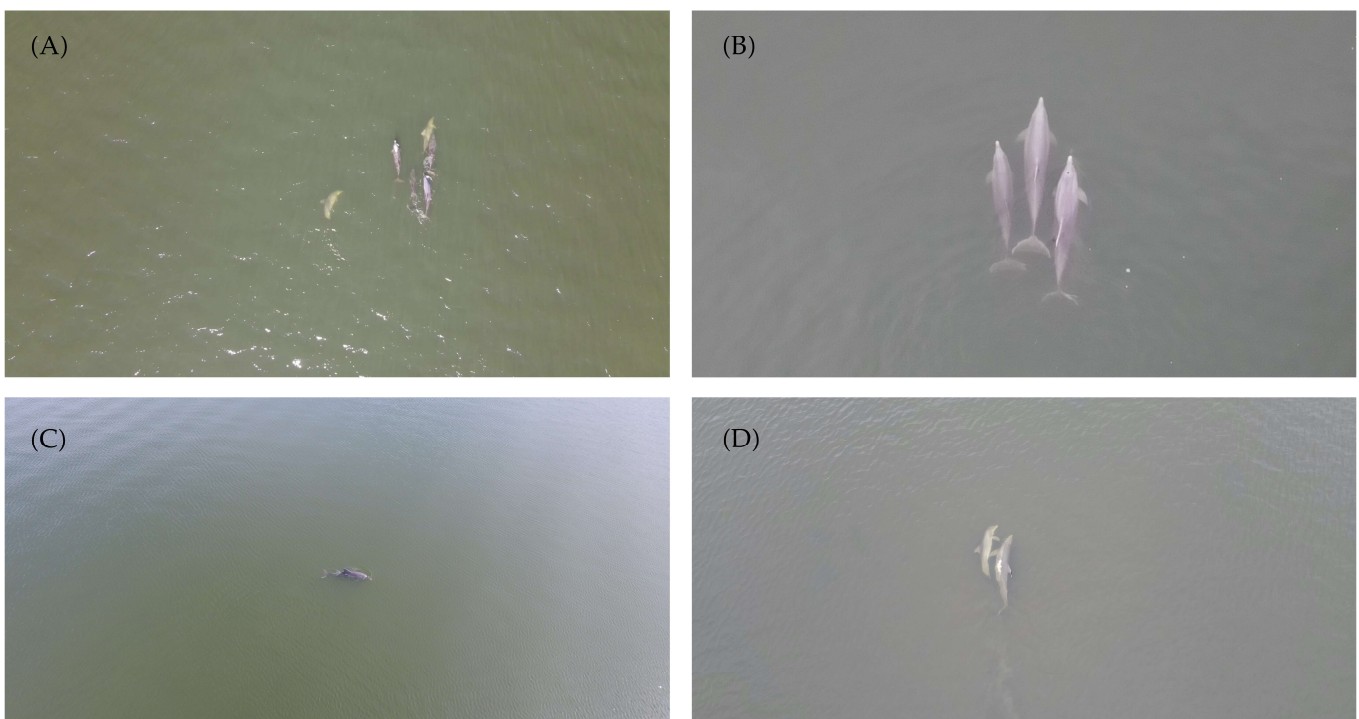

**Figure 3.** Example still images of bottlenose dolphins and water quality taken at various altitudes from UAS surveys: (**A**) 6 dolphins in the Mid-Cooper River, (**B**) 3 dolphins in the Harbor–Cooper River Confluence, (**C**) 1 dolphin in the Folly River, and (**D**) a mom and calf pair in the UCR Site 1.

**Table 2.** Percentage of flights w/dolphin detections, total number of dolphin detections, total dolphins, total mom/calf pairs, and group size estimates from unoccupied aerial system (UAS) surveys across all study sites (*n* = 8) within the Charleston Estuary System (CES).

| Study Site | Percentage of Flights w/Dolphins | Total Detections | Total Dolphins | Mom/Calf Pairs | Density (Total Dolphins/km²) | Mean (±SD) Group Size Per Detection |
|---|---|---|---|---|---|---|
| Folly River (1) | 54.5 | 27 | 49 | 8 | 37.7 | 3.27 (3.26) |
| Harbor–CR Confluence (2) | 49.1 | 34 | 70 | 7 | 31.8 | 2.59 (2.10) |
| Mid-Cooper River (3) | 34.7 | 23 | 48 | 2 | 20.9 | 2.53 (1.95) |
| Upper-Cooper River Site 1 (4) | 14.6 | 7 | 17 | 2 | 6.29 | 3.00 (1.83) |
| Upper-Cooper River Site 2 (5) | 9.4 | 3 | 5 | 2 | 3.57 | 2.33 (1.53) |
| UCR–Back River Confluence (6) | 0 | 0 | 0 | 0 | 0 | 0 |
| Back River Site 1 (7) | 0 | 0 | 0 | 0 | 0 | 0 |
| Back River Site 2 (8) | 0 | 0 | 0 | 0 | 0 | 0 |
| Overall | 26.4 | 94 | 189 | 21 | - | - |

### 3.3. Dolphin Distribution across Seasons

Dolphin sightings were further summarized by season (Figure 5). Dolphins per day did not differ across seasons ($\chi 2$ = 0.475, *df* = 3, *p* > 0.05), though the number of dolphins observed was lowest during winter months (*n* = 31), increased in spring (*n* = 39), and was highest in summer (*n* = 48) and fall (*n* = 48). Additional surveys conducted in January 2022 detected 23 animals. Dolphin occupancy in low-salinity sites varied greatly with season, with higher numbers observed during the summer (*n* = 16), and few in the spring (*n* = 1) and fall (*n* = 5). No dolphins were observed in low-salinity habitats during the winters of 2021/2022.

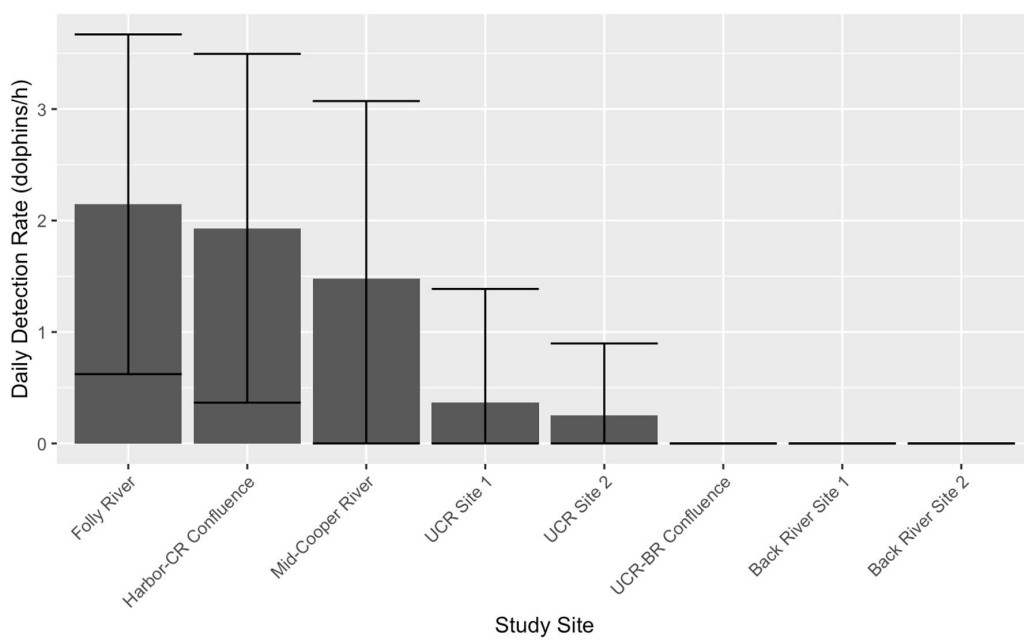

**Figure 4.** Mean (±SD) daily detection rate (number of dolphins per minute of flight time) from unoccupied aerial system (UAS) surveys across each study site (*n* = 8) in the Charleston Estuary System (CES).

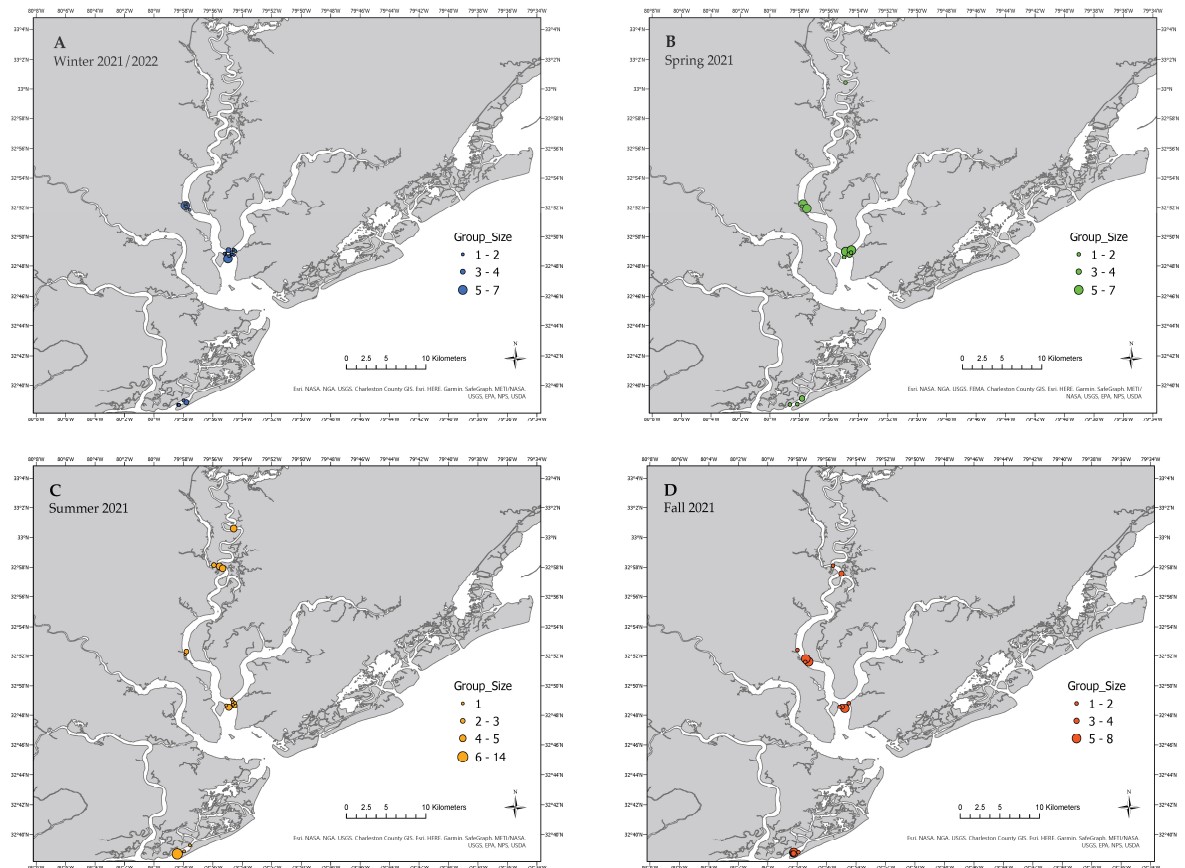

**Figure 5.** Dolphin sightings by season across all study sites (*n* = 8) within the Charleston Estuary System (CES): (**A**) winter (December–February), (**B**) spring (March–May), (**C**) summer (June–August), and (**D**) fall (September–November). Size of the circles represents group size for that sighting.

### 3.4. Dolphin Distribution in Relation to Environmental Variables

The mean salinity (ppt) from the days in which UAS surveys were conducted were highest at the Folly River (25.0 ppt), and decreased moving farther up the system to the Harbor–Cooper River Confluence (21.3 ppt), Mid-Cooper River (14.3 ppt), UCR Site 1 and Site 2 (1.89 ppt), UCR–Back River Confluence (0.044 ppt), and Back River Site 1 and Site 2 (0.045 ppt). The mean water temperature (°C) was similar across sites (Folly River, 20.2 °C; Harbor–Cooper River Confluence, 19.9 °C; Mid-Cooper River, 20.6 °C; UCR Site 1 and Site 2, 18.8 °C; UCR–Back River Confluence, 19.6 °C; Back River Site 1 and Site 2, 18.5 °C). Dolphins were observed in the Folly River and the Harbor–Cooper River Confluence sites year-round, where the mean salinity was above 20 ppt. In the Mid-Cooper River, the mean salinity across all surveys was below the threshold of what is considered high salinity (15 ppt); however, this area does still experience extended periods of high salinity due to tidal influences and has an overall station mean of 15 ppt. Throughout the year, dolphins were observed at this site across a range of salinities (8.65–22.4 ppt; Figure 6), but were more common with lower water temperatures in the spring, fall, and winter. In the UCR Sites 1 and 2, all dolphin sightings occurred when water temperatures were highest, during the late spring–early fall (Figure 6).

Across all eight study sites, if water temperature was held constant, the odds of a dolphin being present increased by 1.13 (95% CI [1.09, 1.16]) for each unit increase (ppt) in salinity. By holding salinity constant, the odds of a dolphin being present increased by 1.03 (95% CI [0.98, 1.08]) for each unit increase (°C) in water temperature. Only looking at the two upper estuarine study sites in which dolphins were sighted (UCR Sites 1 and 2), water temperature appears to be a more significant predictor of dolphin presence compared to salinity. The odds of having a dolphin present increases, holding other factors constant, for a one-unit increase in water temperature by 1.43 (95% CI [1.11, 1.84]). However, due to a small sample of surveys with dolphin sightings at these sites, the interpretation of these results remains limited.

Salinity ($p < 0.001$) and water temperature ($p < 0.01$) in the part of the negative binomial regression model (count model) are both predictors of dolphin abundance. A one-unit increase in salinity (ppt) increased dolphin abundance by 1.05 (95% CI [1.02, 1.07] while holding all other variables in the model constant.

Thus, this model suggested that, with higher salinities, there would be a higher number of dolphins. A one-unit increase in water temperature (°C) increased dolphin abundance by 1.04 (95% CI [1.01, 1.08], while holding other variables constant. Thus, this model also suggested that, with a higher water temperature, there would be a higher number of dolphins.

Salinity in the part of the logit model (zero-inflated model) is a predictor of excessive zeros ($p < 0.001$). With a one-unit increase in salinity (ppt), the estimated odds of observing an excess zero would decrease by 1.12 (95% CI [1.07, 1.16], holding all other variables constant. Thus, the lower the salinity, the less likely it is to observe a dolphin during a UAS flight.

### 3.5. Prey Analysis

The CPUE of 11 prey species were plotted by season for 2021 (Figure 7). Spring had the overall highest CPUE (119 fish/15 min), while winter (57 fish/15 min), summer (41 fish/15 min), and fall (40 fish/15 min) were lower. The type of prey species caught varied by season. Striped mullet, southern flounder, red drum, and American eel were caught year-round. Spot and pinfish were caught from spring–fall, while silver perch was only caught in the summer, bay anchovy was only caught in the winter, and Atlantic croaker was only caught in the spring.

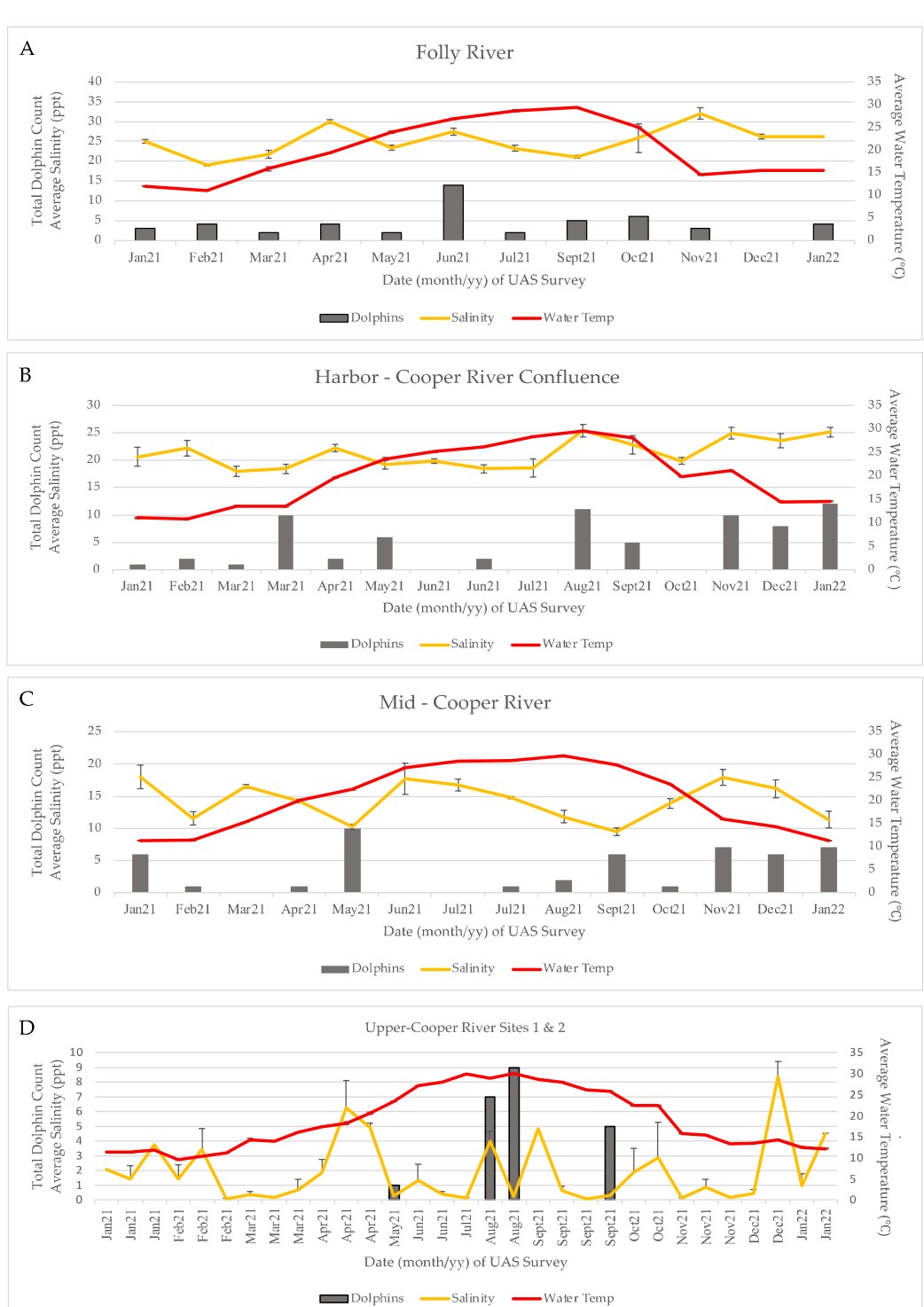

**Figure 6.** Total daily counts of dolphins (gray bar), mean daily salinity (parts per thousand; ppt; yellow line), and mean daily temperature (°C; red line) for study sites (*n* = 5) within the Charleston Estuary System (CES) where dolphins were detected: (**A**) Folly River, (**B**) Harbor–Cooper River Confluence, (**C**) Mid-Cooper River, and (**D**) Upper-Cooper River Sites 1 and 2.

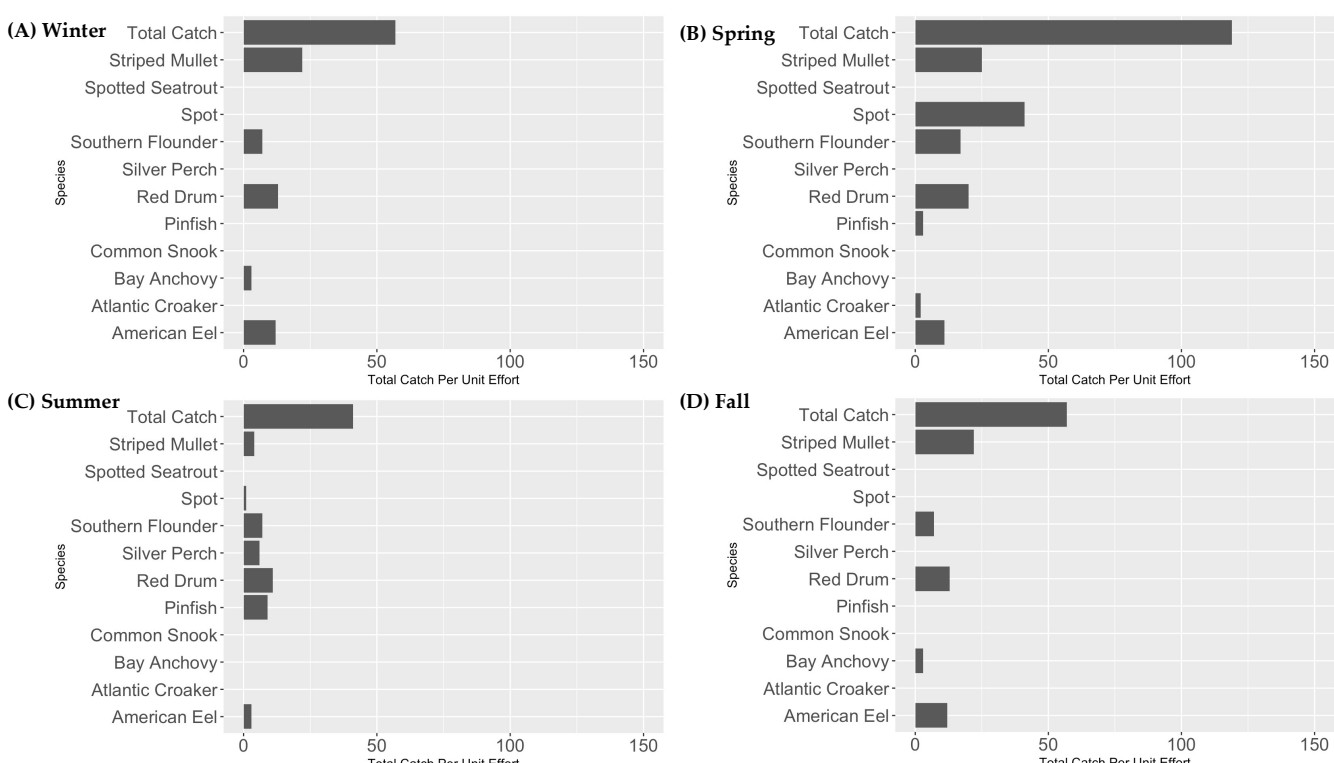

**Figure 7.** Summed total catch of all common dolphin prey species (*n* = 11; first bar) and total catch of each individual species from SC Department of Natural Resources electrofishing surveys in the Upper-Cooper River during 2021 according to season: (**A**) winter, (**B**) spring, (**C**) summer, (**D**) fall.

## 4. Discussion

### 4.1. Bottlenose Dolphin Distribution in the Charleston Estuary System (CES)

Results from this study indicate that bottlenose dolphins can be found within the CES year-round, but fine-scale distribution patterns can vary depending on season, salinity, and water temperature. Through the course of this study, dolphins were observed under a wide range of salinities (0–33 ppt) and in moderate-temperature waters (8.9–31 °C). In high-salinity regions (>15 ppt), dolphins were present year-round; however, in low-salinity regions (<15 ppt), dolphins were observed between May and September when water temperatures were warmest. Overall, salinity was a more significant predictor of dolphin presence than water temperature. However, higher values of both factors led to greater odds of dolphins being present and greater numbers of dolphins present.

#### 4.1.1. Distribution across High-Salinity Sites

Dolphins were most frequently observed in the Folly River, Harbor–Cooper River Confluence, and Mid-Cooper River, where mean annual salinity and water temperature were highest. The Folly River is a coastal tributary and both resident dolphins of the CESS and coastal transient dolphins of the Western North Atlantic South Carolina/Georgia Coastal Stock could be using this area for foraging, socializing, or as a nursery ground, which could contribute to frequent detections [66,67]. This supports Bouchillon et al.'s [22] findings where dolphins were more concentrated near the Charleston coast in the summer months. Dolphins were also frequently sighted in the Harbor–Cooper River Confluence, which was previously identified as a core use area for resident dolphins and may represent an important mixing region between males and females [22]. Similarly, Tribble et al. [68] found high levels of acoustic detections of dolphins in this region, providing further support for this area being a preferred habitat. In the Mid-Cooper River, dolphins were sighted year-round, but were more frequent in winter and spring. The higher number of dolphins in this area during the winter months may be due to dolphins following the distribution of

common prey species including red drum, southern flounder, and spotted sea trout [69]. The presence of dolphins year-round at these three sites, however, supports Bouchillon et al.'s [22] findings that these are preferred habitats for dolphins in the Charleston area and should be considered in any development of protected areas within the CES [22,70].

4.1.2. Distribution across Low-Salinity Sites

Salinity was found to be a significant predictor of dolphin presence across study sites with only a few dolphin detections in low-salinity (<15 ppt) habitats. While the Mid-Cooper River experiences fluctuations in salinity up to 30 ppt, dolphins were observed there on five separate days (Feb–Sept) when salinity levels were between 9 and 14 ppt. Dolphins were encountered in the UCR (Sites 1 and 2) ten times on four separate days (May–Sept) when salinity levels were <5 ppt. In this study, the farthest upriver that dolphins were detected was at the UCR Site 2, where mean annual salinity levels were <2 ppt.

Since 2001, there have been five documented strandings of dolphins in the UCR and six documented strandings of dolphins in the Back River. The Back River is completely freshwater and used to flow freely into the Cooper River until a road was constructed in the 1960's and can now only be accessed by dolphins from a canal to the north. At least four rescue attempts have been made in the Back River when dolphins reached the diked road (Back River Site 2) and did not return north to gain access back to the Cooper River [30]. While stranding history indicates dolphins have previously traveled beyond this site and down into the Back River [30], no dolphins were detected during UAS surveys in the UCR/Back River Confluence, Back River Site 1, or Back River Site 2. This suggests that the dolphins' preferred range may have an upper limit in the Cooper River, with only rare instances of dolphins traveling farther out-of-habitat into the Back River.

Salinity is an abiotic factor believed to heavily influence dolphin distribution [5,71]. Dolphins are efficient osmoregulators and have adapted physiological mechanisms to conserve freshwater and avoid dehydration in hyperosmotic environments. Water flux across a dolphin's skin is dependent on the osmotic gradient of the environment, so dolphins in a hypoosmotic environment will experience a net gain in fresh water [72,73]. While estuarine dolphins can tolerate low salinity, prolonged exposure to freshwater can lead to skin lesions, osmotic imbalance, changes in pathophysiology, increased susceptibility to disease, and mortality [13,32–34,74]. Exposure to waters <20 ppt for as short a duration as 24 h can result in mild serum electrolyte changes [14]. Visually observable skin lesions may require exposure to <10 ppt for days to weeks. However, these lesions and physiological effects associated with low salinity exposure can resolve if/when the individuals return to more saline waters [75]. In Barataria Bay, LA, dolphins were found to frequently use areas with salinities >11 ppt, to sometimes use areas for short periods with salinities near 8 ppt, and to typically avoid waters with salinities <5 ppt [5]. However, in Lake Pontchartrain, LA, a large group of dolphins survived over 2.5 years in waters where mean salinity was 4.8 ppt, although most of these dolphins developed severe skin lesions and likely died as a result [34]. Other studies found that, despite the potential for negative health implications, dolphins did not move in association with salinity changes and did not avoid low-salinity waters even though they could have moved into higher salinity waters [5,76]. It is unknown whether dolphins can sense salinity concentrations, but they could potentially use environmental cues as proxies for salinity, such as the presence of certain fish species, buoyancy, or water turbidity [5]. In this study, surveys were not conducted daily at the same location; therefore, it was not possible to determine whether dolphins remained in those sites for extended periods of time or traveled back and forth into more saline waters. However, in August/September 2022, two dolphins were observed at the Back River Site 2 and remained there for more than 30 days before mortality occurred (personal observation). Future studies in this region should incorporate systematic photo ID surveys to better understand whether it is only a small number of dolphins that more frequently use the upper regions of the Cooper River.

If salinity has increased over time in the UCR, potentially due to sea level rise and climate change, these upper estuarine habitats may be more attractive to dolphins to travel to and spend prolonged periods of time in. However, this region has not experienced a marked increase in salinity over time, but does experience periodic fluctuations where it can reach as high as 10–12 ppt at times (see Supplementary Text S2 for details) [77,78]. While salinity is an important factor, the results from this study suggest that water temperature is a more significant predictor of dolphin presence in this region, with a greater chance of dolphins being present with warmer temperatures regardless of the salinity concentration. In this region, all dolphin sightings occurred during the late spring–early fall, peaking in the summer. This coincides with the timing of previous strandings of dolphins in this area, where the majority (82%) occurred during the summer months [30]. Similarly, in Barataria Bay, Louisiana, satellite telemetry revealed that dolphins tended to use northern parts of the bay, where salinities were <5 ppt, during summer months [6]. Further, it was shown dolphins did not move to avoid low-salinity conditions for the duration of the telemetry tags, suggesting that other factors drive their movements, such as predator/prey availability, mating opportunities, sexual behavior, and/or social organization [6]. Similar patterns were also observed in the upper Galveston Bay, TX (UGB), an area also considered a sub-optimal dolphin habitat in terms of low salinity levels [71]. Several studies indicated that dolphins are present in the UGB year-round, with encounter rates rising during months with the warmest water temperatures [71,79,80].

The influence of water temperature on dolphin distribution may be more indirectly related through prey availability. One previous hypothesis explaining why dolphins may be traveling into low-salinity habitats in the UCR and Back River is that they are following additional prey resources [30]. Direct observations of dolphins chasing and catching fish were made from UAS surveys, with at least one prey species identified as a school of striped mullet [81]. Stomach content analyses from dolphins that were previously stranded in the UCR and Back Rivers identified several species including Atlantic croaker, silver perch, spotted seatrout, bay anchovy, snook, and spot as prey items [30]. Additional species present in the UCR that have been identified as prey for dolphins in South Carolina include American eel, pinfish, red drum, southern flounder, and striped mullet [65]. During 2021, the mean CPUE was highest overall in the spring and generally comparable in winter, summer, and fall. When dolphins were detected in the UCR, a diversity of common prey species were present including American eel, Atlantic menhaden, southern flounder, spot, striped mullet, inland silverside, largemouth bass, pinfish, red drum, and silver perch, suggesting ample food resources in this region.

In estuaries, prey migration in relation to spawning cycles and water temperature is likely to influence dolphin distribution. For example, dolphins in Sarasota Bay, Florida, may shift their distribution toward the Gulf of Mexico in cooler months when mullet migrate into coastal waters to spawn [82]. It is similarly hypothesized that dolphins return to UGB with rising water temperatures due to food availability, including the presence of mullet [71]. Mullet (*Mugil* sp.) has been previously described as a preferred prey species for dolphins in several regions, including South Carolina estuaries [10,65,83]. Striped mullet spawns offshore in South Carolina between October and April, peaking in December through February. The eggs and larvae move inshore, and the juveniles use estuaries as nursery areas [84]. They are commonly encountered year-round, with electrofishing surveys catching newly settled fish (<2 in) during winter and early spring. At this size, mullet are likely too small to be effective prey for dolphins. By summer and fall, mullet are larger (8–20 in) and more suitable prey items [81]. It is possible that the greater number of dolphins in the UCR during the summer and fall is from dolphins pursuing the larger mullet. The CPUE of mullet in the UCR has varied over time, however, with declines in recent years occurring throughout the CES [81,85]. While they may still be an important driving force behind dolphin distribution in the UCR, it is possible that dolphins are not attempting to exploit large numbers of additional prey upriver, but instead are being forced to find food upriver due to low abundances in the lower estuaries. More research is needed

to determine foraging hotspots of dolphins in the CES and if prey availability in relation to water temperature is a significant factor driving dolphin distribution.

While salinity, water temperature, and prey are likely important environmental factors in determining dolphin distribution, it is likely there are other factors that can influence distribution, such as tidal state, dissolved oxygen, bottom topography, and depth. Additionally, more surveys should be conducted to collect distribution data over multiple years to increase sample size.

*4.2. Benefits and Limitations of UAS Methods to Study Bottlenose Dolphins*

Utilizing a UAS to survey dolphins within the CES presented several benefits and limitations. Two out-of-box DJI quadcopters were utilized which were low-cost, easy to maneuver, and provided high-quality footage of bottlenose dolphins. This study was successful in detecting dolphins via UAS to understand distribution patterns. Dolphins were detected in 26% of UAS flights across study sites in the CES. There were only three occasions of dolphins being detected during the post-survey review of video footage that were not initially seen while conducting the survey.

Launching the UAS from land compared to launching from a vessel had several limitations. Dorsal fin images of dolphins were not able to be collected as the dolphins were typically too far away from land and it was not possible obtain clear dorsal fin photos from the UAS that could be used for photo identification. It also limited the amount of area that could be surveyed at each study site, due to maintaining VLOS and a strong connection between the UAS and remote control. Additionally, due to the narrow river widths of some study sites, some transects overlapped in certain areas. However, if dolphins were sighted travelling unidirectionally during one flight, waiting 30 min before launching the UAS again was typically sufficient time for those animals to travel beyond the range of the study site and not be re-counted. However, there were instances of dolphins staying within the study site for an extended period, either foraging or socializing. It was usually possible to know whether the same dolphins were sighted again during subsequent flights based on behavior, group size/composition, and the general direction they were heading in. There were likely a few instances where the same dolphin was counted twice unknowingly or an animal that was believed to be the same animal and not counted again was in fact a new animal. Therefore, conducting UAS surveys from land-based home points can lead to some bias in estimating total dolphin counts.

This study was successful in obtaining group count estimates, with group sizes varying from 1 to 14 animals. While marine mammals have traditionally been surveyed via vessel-based surveys not using a UAS, limited observations from the horizontal perspective can lead to low-bias estimates of group counts with missing individuals [86,87]. Several other studies have shown support for using UAS for assessing the populations of marine mammals as it can eliminate certain observer bias [88]. Oliveira-da-Costa et al. [44] used a UAS to detect two Amazon dolphin species (*Sotalia fluviatilis* and *Inia geoffrensis*) and found these surveys provided higher accuracy than human observers in counting individuals in a group. Similarly, Fettermann de Oliveria [48] compared UAS-derived estimates of group sizes to vessel-based estimates of bottlenose dolphins off the Great Barrier Island, NZ, and determined that UAS-derived observations detected higher counts of dolphins, demonstrating that UAS surveys can improve the accuracy of population counts for small cetaceans.

In this study, the UAS was flown at 30 m in altitude in accordance with NMFS Permit #21938-03 to minimize the disturbance to dolphins. During brief descents down to altitudes no lower than 9 m, there were a few instances of dolphins turning on their side, which appeared to be a potential curious response to the UAS. However, no other signs of disturbance (i.e., rapid or erratic movements, shorter surface time, or change in behavior) were observed among the dolphins as a result of the operation of the UAS. This is similar to results from studies assessing the impacts of UAS on small cetaceans [89,90]. The video footage of dolphins collected in this study also allowed for other detailed observations

of behavior. Future work will include behavioral assessment of the footage to establish baseline habitat use patterns across study sites.

While the absence of dolphins altogether in three low-salinity sites likely contributes to a low overall detection rate, other environmental variables may have affected detectability, including turbidity and light conditions. While coastal cetaceans may be clearly visible from above several meters deep in clear shallow water habitats, in turbid areas with muddy bottom habitats, dolphins may not always be visible below the surface. While turbidity measurements were not directly taken in this study as most surveys were conducted from land, water clarity was generally assessed through video observations. Turbidity appeared highest during summer months, leading to some instances where dolphins were only visible when they surfaced. In contrast, on days when the water was clearer, dolphins were visible below the surface, making it easier to follow them with the UAS. Therefore, the high turbidity of estuarine habitats can result in reduced detections and sighting durations, and subsequent inaccuracies in estimating group size and identifying new individuals. Ramos [91] also found it challenging to reliably distinguish different individual bottlenose dolphins over time in aerial videos from UAS surveys, which prevented observations in which animals could be reliably tracked for more than a few surfacings, particularly in groups of three or more animals. Abundant sunshine also caused challenges with visibility and glare, which may have led to missing dolphins. The effect of cloud cover on detection rates was not directly assessed in this study.

## 5. Conclusions

This study utilized UAS surveys to assess the distribution of bottlenose dolphins across different regions in the Charleston Estuary System in terms of salinity, water temperature, seasonality, and prey availability as well as to discuss the benefits and limitations of using UAS methodologies to survey dolphins in a complex, turbid estuary. Dolphins were detected year-round across high-salinity sites and infrequently detected in low-salinity sites. Dolphins were only detected in low-salinity habitats when water temperatures were warmest (May–September). Dolphin movement into the upper estuarine waters of the Cooper River during warmer temperatures may be in response to prey distribution, with mullet migration being a possible factor.

Regardless as to why, if dolphins remain in these low-salinity habitats for extended periods, negative health consequences from prolonged freshwater exposure are possible. The continued monitoring of dolphins in these non-optimal habitats could lead to the further understanding of potential shifts in distribution of which conservation or management plans may need to be developed, especially as climate conditions continue to change. Increasingly warm water temperatures may lead to further shifts in prey distribution, potentially forcing more estuarine dolphins to utilize upper riverine habitats to find food. Sea level rise may also lead dolphins to alter their distribution as salinity changes throughout the estuary. Additional methods to continue monitoring dolphins in non-optimal habitats could include deploying acoustic devices for passive acoustic monitoring, as well as incorporating citizen science efforts. Allowing the public (e.g., local fishermen and boaters) to document sightings of dolphins in the UCR and Back River can help provide additional data and information about when and where dolphins are in those regions. Future studies building upon this UAS study will provide a more comprehensive and long-term understanding of how biotic (predator/prey) and abiotic (tide, salinity, water temperature, DO, etc.) factors affect dolphin distribution. While this study recognizes several limitations, it adds to a growing body of literature supporting the use of UASs for conducting aerial surveys of bottlenose dolphins in complex estuaries.

**Supplementary Materials:** The following supporting information can be downloaded at: https://www.mdpi.com/article/10.3390/drones7120689/s1, Text S1: Historical (2000–2021) salinity data collection in the Upper-Cooper River; Text S2: Looking at trends in historical salinity concentrations in the Upper-Cooper River from 2000 to 2021; Figure S1: Coastal Salinity Index (CSI) at U.S. Geological Survey (USGS) Station 02172053 Cooper R at Mobay, N Charleston, South Carolina, from 2000 to 2021, located near Upper-Cooper River Sites 1 and 2.

**Author Contributions:** Conceptualization, N.P., W.M. and N.L.; Data curation, N.P. and J.B.; Formal analysis, N.P.; Investigation, N.P.; Methodology, N.P., W.M., N.L. and B.B.; Project administration, W.M. and N.L.; Resources, W.M. and N.L.; Supervision, W.M. and N.L.; Validation, W.M.; Visualization, N.P.; Writing—original draft, N.P.; Writing—review and editing, W.M., N.L., B.B., and J.B. All authors have read and agreed to the published version of the manuscript.

**Funding:** This research received no external funding.

**Institutional Review Board Statement:** This study was conducted in accordance with National Marine Fisheries Permit 21938-03. A review by the College of Charleston Institutional Animal Care and Use Committee (IACUC) determined that the study qualified as a field observational study and that no IACUC was required.

**Informed Consent Statement:** Not applicable.

**Data Availability Statement:** The dataset created during the current study is not publicly available, but is available from the corresponding author on reasonable request.

**Acknowledgments:** Thank you to Michael Wiser for the UAS assistance and acting as a visual observer and to Colin Perkins-Taylor for also acting as a visual observer.

**Conflicts of Interest:** The authors declare no conflict of interest.

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
