# Peer review of "Using Unoccupied Aerial Systems (UASs) to Determine the Distribution Patterns of Tamanend’s Bottlenose Dolphins (Tursiops erebennus) across Varying Salinities in Charleston, South Carolina"

_drones, doi:10.3390/drones7120689_

Round 1

Reviewer 1 Report

Comments and Suggestions for Authors

The authors have investigated the use of UAS to assess bottlenose dolphin distribution across an estuarine system. I find this study very interesting and a great contribution to the knowledge base on UAS methodology. My first impression was that this study has a lot of potential, but the lack of standardization in the sampling process is of concern. The article is well written and well organized, and the authors do a good job in synthesizing the available literature and answering relevant questions about the topic field. The analysis of methodology and terminology for marine mammal surveys is not as developed as I would expect to see in an academic article on this topic and I would therefore suggest the authors to revise their material to showcase the current knowledge. There are quite a few shortcomings that should be addressed prior to publication, and a significant revision would be advised.

Overall, the introduction is very well written, but the methodology is unclear, and I would advise the authors to add reasoning behind the sampling approaches so that the section is more solid. The terminology is also a bit confusing and I am unsure of the validity of the selected analytical approaches. Further, I would also advise the authors to present the survey lines/transects to showcase the effort at each site.

The authors do a good job in estimating detection rate, but this metric is then lost and no longer used in the rest of the assessment of variation in dolphin occurrence either in relation to environmental features or seasonally. This would be an important aspect to consider especially when the same on-effort duration is not done at the resolution of the data (e.g., per season or per day or even per site) and could bias the results. I see that there is about twice as many flights in Harbor CR Confluence than in Back river site 2, so it would be definitely important to not just consider absolute values when comparing number of animals/detections across sites. Plus, the article would really benefit from some boxplots to show the variation in detection rate per season colored (or faceted) by site, so that it could be possible to discriminate associations between site (and season) combined rather than separately. Finally, can we really trust such small sample sizes in one year of data when looking at comparisons between seasons? I would advise the authors to not focus on seasonal differences as an own section but rather the more “pure” environmental variations that occur at each site throughout the sampling period. Given that prey availability was not possible to collect at the same time and location as the UAS surveys, an alternative to consider would be animal behavior. The authors did record video and stopped the transect flights to further assess group size, so there should be good information available on behavior (as also in the discussion L548).

Given the above comments and suggestions, I recommend major revisions of this work for publication at this journal. I suggest that the authors spend more time to expand and describe the methods and results. Please see more of my comments in the attached document to help improve the quality and readability of the manuscript.

All the best with this paper.

Reviewer 2 Report

Comments and Suggestions for Authors

This is an interesting study as it investigates the population of Tamanend’s bottlenose dolphins in the CES, along with the effects of biotic and abiotic factors on their presence/absence, relative abundance, and detection rate. Additionally, it introduces a novel use of UAVs for studying dolphins in a turbid, complex estuarine environment. The paper is well-written and thorough, with only minor points to address. Subjectively, it could benefit from some slight shortening of the introduction and discussion. 

There are several self-citings (McFee: reference 1, 27, 33, 35, 65; Levine: reference 22; Balmer: reference 5, 6), but they all appear relevant since this study covers a quite narrow field. However, I am not sufficiently well versed in this branch of the literature to tell if there are more appropriate references.

Line 104: The word “Review” is in the references. Maybe a placeholder to be deleted?

Figure 3: Please visualize which sites are significantly different, as well as adding it in the figure text.

Line 391: Remove underline in (ºC)

Line 549, 571, 574, reference 80, Ballenger. Unsure if this is the correct way to cite personal communication/knowledge of a co-author in a MDPI journal?

Line 580-584: Repetition. If meant as a summation, please make it clear.

Reviewer 3 Report

Comments and Suggestions for Authors

Review of Drones submission titled: “Using Unmanned Aerial Systems (UAS) to Determine Distribution Patterns of Tamanend’s Bottlenose Dolphins (Tursiops erebennus) Across Varying Salinities in Charleston, South Carolina”

Overall impression

In this study, Principe et al use commercial unmanned aerial systems to evaluate the distribution patterns of Tamanend’s bottlenose dolphins in the Charleston Estuarine System Stock in South Carolina, USA. The authors conducted surveys with unmanned aerial systems from shore in search of dolphins. Their aim was to assess the distribution patterns of dolphins in the estuary and the ecological influences of their presence, distribution, and abundance, in particular, salinity.  They were able to detect dolphin presence throughout the year most in high salinity areas, and less detections in lower salinity waters associated with seasonal increases in water temperatures.

The study is well done and the manuscript is well-written. There are some questions that requiring addressing and some points to clarify throughout.

One methodological concern I have is related to needing to know more about the specific flights patterns and area covered. It’s unclear in the methods what kinds of transects were flown and over where. Understanding the design of the flights is critical to being able to replicate this study, thus, more information should be provided to clarify this. It will also help downstream in the manuscript where some of the results and discussion are difficult to interpret absent understanding of flight parameters.

The discussion gets a bit muddled in the salinity part so I would work on restructuring that section a bit.

Not essential, but it was recommended to me in multiple forums that unoccupied aerial systems or unoccupied aircraft systems is a more gender neutral terms preferred over unmanned.

Otherwise, great study! Congratulations to the authors for the great job.

Reviewer comments:

Abstract

Somewhere in the abstract, say what state and country. For an international audience, it might be hard to pin where this is. At the end of the first sentence would be a good place.

Keywords

Better to not use words that are in the title

Introduction

Line 118 – maybe reword to state the assessment of dolphin distribution as the main aim, and the testing of the UAS methods as secondary.

Line 130 – I do not think this should be specific hypothesis as while I agree you are able to show UAS is effective for your goals in this context, its not something explicitly tested in comparison to some other method directly.

I would say this is more of your overarching goals, illustrated through your study and its findings.

Line 136 – good place to reference Figure 1 for context

Methods

Line 177 – Can you please provide some graphic depiction of these transect lines and/or go into depth as to what kind of transects they were. Given the convoluted winding nature of the estuary from the map, its important to know how this was handle in term so area covered. Also how were location of home points selected? What were the range of distances flown and area covered per study site?

Similarly, can you provide some information on how much area coverage these flights gave you? What I mean is, how wide was your image frame and pixel dimensions in real life? You flew at 30 m, correct? If this is the case, then your FOV may not cover a very large area and understanding the relationship between the area you were able to cover and the dimensions of your FOV is important is interpreting the effectives of this method for surveying estuaries.

Line 214 – How were flight records used? What was the means of accessing them and extracting the information. In providing your innovative application for this tool, its important to also accurately transmit the steps used to get there that may not be intuitive or obvious to the reader or future practitioners.

Line 248 – To assess if the number

Results

Line 338 – Can you clarify what you mean by this? Also having some understanding of this in the methods would be important because it is a limitation of your method that will have to be accounted for in other studies.

Line 362 – the end of the sentence is redundant as you states number of detections in the beginning of the sentence.

Line 365 – be explicit here instead of referencing the previous test.

Line 377 – “Dolphins per day” is confusing. Is this number of dolphin detections or raw number of dolphins? If it’s the latter, maybe say “total number of individual dolphins detected” or some similar wording

Line 379 – double check this, but I am pretty sure the seasons should be capitalized

Line 383 – habitats

Line 413 – the odds of a dolphin being present

Line 418 – were both predictors

Line 428 – should this be “the lower the salinity, the less likely it is”?

Discussion

Should 4.1 be italicized and the other subheadings not?

Line 475 – I recommend coming up with a better topic sentence than this, especially given the subheading title. Its not immediately clear in the first few sentences why what you are saying is relevant. Something explicitly summarizing the relationship you found between dolphin distribution and salinity would be better.

Line 497 – its not immediately clear how the preceding sentences relate to the next statements. Reword somehow to make a smoother transition.

Line 500 – given where you go in this paragraph, I recommend restructuring these two paragraphs a bit as they feel out of order. This information on their exposure to freshwater and higher risk of mortality would be better placed closer to the statement regarding the dolphins dying in the Back River.

Line 526 – delete the extra space before However

Line 631 – suggest referencing work previously reporting these behaviors:

Ramos, E. A., Maloney, B., Magnasco, M. O., & Reiss, D. (2018). Bottlenose dolphins and Antillean manatees respond to small multi-rotor unmanned aerial systems. Frontiers in Marine Science, 5, 316.

Fettermann, T., Fiori, L., Bader, M., Doshi, A., Breen, D., Stockin, K. A., & Bollard, B. (2019). Behaviour reactions of bottlenose dolphins (Tursiops truncatus) to multirotor Unmanned Aerial Vehicles (UAVs). Scientific Reports, 9(1), 8558.

Giles, A. B., Butcher, P. A., Colefax, A. P., Pagendam, D. E., Mayjor, M., & Kelaher, B. P. (2021). Responses of bottlenose dolphins (Tursiops spp.) to small drones. Aquatic conservation: marine and freshwater ecosystems, 31(3), 677-684.

I would remove the paragraph in like 645 to almost 670. This is an interesting observation, but new results shouldn’t be reported in the discussion. It is also tangential from your study, while connected. I recommend mentioning only briefly these possibilities and then leaving the other ideas for another study.

I suggest breaking up the conclusions and shortening. Much of this has already been stated, and does not need to be reiterated in such detail at the end of the paper. Typically, a conclusions paragraph gives you the main take home points and discusses future prospects and implications. If you feel you need to retain some of this information written in this paragraph, I recommend moving it to prior to the conclusions, and keeping the conclusions succinct.

Tables and Figures

Figure 2

Could you include some more images to give an idea of what the range of dolphin detections looked like?

Figure 3

Total Number Per Hour…

Figure 4

Seasons should be capitalized.

Reduce the number of latitudes and longitude ticks and increase the number size so it is similar to text size in manuscript, and legible when view in this format.

It would be useful in this figure or figure 1, to place an inset map showing a zoomed version of how you covered individual study sites as exemplars.

Put the labels at the top left corner to be consistent with your other figures and the general conventions

Figure 5

In d, the 10 is obscured by the labeling

Rotate the 2nd y axis so that the text is opposite from the other axis. More difficult to read this way

It may be useful to make the dates in a format more common with international standards ex: yyyy-mm-dd

Can you adjust the axes so that they are all rotated to the same degree?

Table 1

All the values in Total Flight Time should have two decimal places to be consistent

Comments on the Quality of English Language

The english is fine and just needs some minor fixes throughout.

Reviewer 4 Report

Comments and Suggestions for Authors

The work is not novel with respect to the use of drones in the study of marine mammals, it is a very simple work on cetacean ecology that provides more knowledge about the biology of this coastal species in the region of the South Atlantic Bight.

For me, the work is methodologically correct and presents new information on marine resources; so I consider that it can be published with some considerations that I leave below, and that are in the attached pdf file.

In the discussion about the seasonality in the presence of the species in the estuary, the authors should be include the seasonal changes in the ocean productivity of the South Atlantic Bight region, which is greater in winter and lower in the summer; this contrast in the environment must force the movement of some of dolphin prey to the coast or to the open sea (https://www.sciencedirect.com/science/article/abs/pii/S0278434397000022; https://www.tandfonline.com/doi/abs/10.1080/01965581.1985.10749480)
